# Peptide nucleic acids can form hairpins and bind RNA-binding proteins

**Yichen Zhong**[1], **Lorna Wilkinson-White**[2], **Esther Zhang**[1], **Biswaranjan Mohanty**[2], **Belinda B. Zhang**[1], **Madeline S. McRae**[1], **Rachel Luo**[1], **Thomas A. Allport**[1], **Anthony P. Duff**[3], **Jennifer Zhao**[1], **Serene El-Kamand**[4], **Mar-Dean Du Plessis**[4], **Liza Cubeddu**[1,4], **Roland Gamsjaeger**[1,4], **Sandro F. Ataide**[1]*, **Ann H. Kwan**[1]*

**1** Currently or formerly at School of Life and Environmental Sciences, The University of Sydney, Sydney, NSW, Australia, **2** Sydney Analytical Core Research Facility, The University of Sydney, Sydney, NSW, Australia, **3** National Deuteration Facility, ANSTO, Lucas Heights, NSW, Australia, **4** School of Science, Western Sydney University, Penrith, NSW, Australia

* ann.kwan@sydney.edu.au (AHK); sandro.ataide@sydney.edu.au (SFA)

**Data Availability Statement:** All relevant data are within the manuscript and its Supporting Information files.

**Funding:** The University of Sydney Drug Discovery Initiative (DDI) seed funding The production of

## Abstract

RNA-binding proteins (RBPs) are a major class of proteins that interact with RNAs to change their fate or function. RBPs and the ribonucleoprotein complexes they constitute are involved in many essential cellular processes. In many cases, the molecular details of RBP: RNA interactions differ between viruses, prokaryotes and eukaryotes, making prokaryotic and viral RBPs good potential drug targets. However, targeting RBPs with small molecules has so far been met with limited success as RNA-binding sites tend to be extended, shallow and dynamic with a mixture of charged, polar and hydrophobic interactions. Here, we show that peptide nucleic acids (PNAs) with nucleic acid-like binding properties and a highly stable peptide-like backbone can be used to target some RBPs. We have designed PNAs to mimic the short RNA stem-loop sequence required for the initiation of prokaryotic signal recognition particle (SRP) assembly, a target for antibiotics development. Using a range of biophysical and biochemical assays, the designed PNAs were demonstrated to fold into a hairpin structure, bind the targeted protein and compete with the native RNA hairpin to inhibit SRP formation. To show the applicability of PNAs against other RBPs, a PNA was also shown to bind Nsp9 from SARS-CoV-2, a protein that exhibits non-sequence-specific RNA binding but preferentially binds hairpin structures. Taken together, our results support that PNAs can be a promising class of compounds for targeting RNA-binding activities in RBPs.

## Introduction

RNA-binding proteins (RBPs) are a major class of proteins with over 2000 members. RBPs participate in diverse cellular functions and have been implicated in cell homeostasis, growth, division, differentiation, and cell fate determination through their interactions with RNA and other proteins [1, 2]. However, despite the abundance and importance of RBPs, delineating the exact role and mechanism of RBP:RNA interactions is challenging due to the lack of specific and robust molecules that target RNA-binding interfaces. If such compounds exist, they

2H13C15N FtsYNG was supported by grant NDF9615 from the National Deuteration Facility, which is partly supported by the National Collaborative Research Infrastructure Strategy – an initiative of the Australian Government. The funders of Drug Discovery Initiative Seed Grant and ANSTO National Dueteration Facility (NDF) Grant had no role in study design, data collection and analysis, decision to publish, or preparation of the manuscript. NDF staff scientist, Dr Anthony Duff, is a co-author and led the production of isotopically labelled FtsY and proofread the manuscript.

**Competing interests:** NO authors have competing interests.

can be used to abrogate RBP:RNA interactions in assays to understand their effect in biology and to assess new putative RBP drug targets. In contrast, many small-molecule antagonists and agonists have been discovered for major protein classes such as enzymes and receptors [3, 4], as well as engineered peptides and proteins (including antibodies) that specifically recognise protein-interaction interfaces [5, 6]. Understanding RBP:RNA interactions would offer new avenues to probe bacterial and viral biology and may yield new antimicrobial strategies, as these interactions often differ substantially in molecular details and component compositions between lower and higher organisms, even in functionally conserved pathways [1, 2]. While some basal RBP:RNA interactions remain universal, such as the use of ribosomal protein and ribosomal RNA (rRNA) interactions for protein synthesis [7], higher level gene control elements typically have notable distinctions between viruses, prokaryotes and eukaryotes [8].

Unlike enzymes with small well-defined active site pockets that can easily be inhibited by small molecules, RNA-binding sites on RBPs are typically large, shallow and dynamic with a mixture of charged, polar and hydrophobic interactions [9], making them hard to target using traditional high-throughput screening approaches with small molecules. Despite the abundance of RBPs with ~1500 members encoded by the human genome [10], the difficulty of targeting is evident in the bias of candidates towards enzymes, with 90% of over 800,000 unique chemical structures interacting with only the top 278 enzymatic or receptor targets [11]. Therefore, the development of RNA mimics that can bind RBPs may help to overcome the lack of success with high-throughput screening strategies and lead to new and easy-to-use tool molecules to target RBPs. These molecules also have the potential to be developed into novel drug leads.

Peptide nucleic acids (PNAs) are compounds developed with nucleic acid-like binding properties and exceptional biological stability that can even surpass cyclic peptides. Discovered by Nielsen *et al.* in 1991, PNAs are synthetic, comprise of nucleobases covalently attached to an achiral polyamide (peptide-like) backbone as opposed to the ribose phosphate backbones of RNA [12]. The lack of phosphate groups on PNAs results in a removal of electrostatic repulsion as a major factor impeding hybridisation and base-pairing, making PNA duplexes stable even in solutions of high ionic strength [12, 13]. The amide backbone makes PNAs completely acid and base stable, and additionally not recognisable by nucleases [14], resulting in a greater chemical and biological stability. PNAs can form parallel and antiparallel duplexes with itself, generating a P-form helical structure with both a deeper and wider major groove compared to RNA and DNA A and B-form helices [15]. PNA applications so far have exploited its ability to hybridise double-stranded DNA or RNA, forming a triplex-invasion complex and eventually displacing one of the strands [16]. Previous studies have only reported uses of PNAs as a gene knockdown agent, as opposed to target RBPs interfaces. Whilst PNAs have been used as RNA-mimicking inhibitors of human telomerase [17], the use of PNAs as RNA-mimicking inhibitors of RBPs has remained unexplored.

We propose that PNAs may represent a suitable class of molecules that can be used to target a subset of RBPs, in particular, where RBPs display shallow, dynamic and transient binding to RNAs over an extended binding interface with low to medium binding affinities (in the μM range). This type of binding is particulary common where RBPs participate in conformational switching mechanisms and bind multiple RNA sites or sequences as in anti-terminators like EutV among others [18]. In many instances, the recognition of sequence and structural features (single stranded, stem-loops and double stranded) are the major contributors to binding of RBPs and ribonucleoprotein (RNP) formation with the negative charges of the RNA backbone playing only a relatively minor role. In addition, PNAs can be engineered to contain glutamates in selected locations to mimic the negative charge in the RNA backbone. We hypothesized one of the dynamic protein-RNA interactions that can be targeted is the signal

recognition particle (SRP) and its SRP receptor (SR), an essential RNP complex responsible for protein sorting in the cell [19]. In prokaryotes, the SR (FtsY) switches between interacting with the tetraloop and distal loop of the 4.5S RNA. The binding to the tetraloop, *i.e.*, a protein: RNA interaction, promotes complex formation with the SRP protein (Ffh), which forms heterodimers with FtsY through the N-terminal helical bundle and GTPase domains (NG domain). In contrast, eukaryotic SRP and SR are composed of multiple proteins and the SRP: SR complex formation is mediated through protein-protein interactions [20]. As the protein sorting pathway is essential for cell survival, selective targeting of the protein-RNA interaction in prokaryotes may enable the development of potent antibiotics [19, 21, 22]. Here we explore whether PNAs containing the tetraloop sequence can mimic the 4.5S RNA tetraloop to inhibit the $FtsY_{NG}$:4.5S RNA interaction. Using a combination of biophysical and biochemical experiments, we have shown that the designed PNAs based on the 4.5S RNA tetraloop can fold into a stemloop structure and bind $FtsY_{NG}$. Using electrophoretic mobility shift assay (EMSA) and microscale thermophoresis (MST) assays, we have shown that the PNAs with the tetraloop sequence can prevent SRP:SR complex formation.

To show the potential applicability of PNAs against other RBPs with a dynamic and transient binding mechanism, we demonstrated that a PNA could also bind to the Nsp9 protein from SARS-CoV-2. Nsp9 is an essential RBP that is known to bind RNA with low to moderate affinity in a non-sequence specific manner [23, 24]. However, more recently Nsp9 has been shown to recognise RNA hairpin structures with a much higher affinity [25]. Taken together, we have demonstrated for the first time that PNAs have the potential to be used in a combination of sequence and structural mimics of RNA moieties to target RBPs and inhibit RNA binding. Furthermore, the high structural, chemical and biological stability of PNAs offer many practical benefits and enable a range of assays under experimental conditions where RNA cannot be used.

## Materials and methods

### *In vitro* transcription and purification of 4.5S RNA

The plasmids for transcribing full length 4.5S and RNA$^{s4.5S}$ (5' TGTTGGTTCTCCCGCAC**GGAA**GTGCCGGGATGTAGCTGGCA 3') are constructed as described in [26]. Briefly, the DNA templates was flanked by a hammerhead ribozyme at the 5′ end and a hepatitis delta virus (HDV) ribozyme at the 3′ end, each of which self-cleave during transcription to yield product RNA with homogeneous termini. A sequence comprising an EcoRI restriction site for cloning and a T7 RNA polymerase promoter was added to the 5′ end of the DNA template, whereas the 3′ end includes a BamHI restriction site for cloning and the terminus of the HDV ribozyme. The entire sequence was then cloned into EcoRI and BamHI sites of pUC19 vector.

Cloned pUC19 plasmids was digested overnight with BamHI-HF (150 U per mg of DNA) at 37˚C. On the next day, the linearised DNA was purified with phenol-chloroform extraction followed by chloroform extraction and then ethanol precipitation. The DNA resolubilized in MilliQ water is used as template (0.5 mg per 10 mL reaction) for *in vitro* transcription by mixing with 40 mM HEPES-KOH, pH 7.5, 40 mM $MgCl_2$, 0.1 mg/mL BSA, 2 mM Spermidine, 40 mM DTT, 7.5 mM of each NTP, 0.01 mg/mL Pyrophosphatase, 0.02 U RiboSafe RNase inhibitor (Bioline, Cat. No.: BIO-65028) and 0.1 mg/mL T7 RNA polymerase (produced in-house). The reaction was incubated at 37˚C for 2 h, followed by another 2 h of incubation at 42˚C.

For ribozyme self-cleavage, the transcription products were replenished with additional 20 mM $MgCl_2$, then incubated at 95˚C for 2 min and immediately on ice for 3 min. Repeat the heating cycles three times in total. After mixing with 2× RNA loading dye (80% (v/v)

formamide, 1× TBE, xylene cyanole FF and bromophenol blue), the sample was heated at 95˚C for 2 min before loading onto a 6% denaturing urea gel consisting of 7 M urea, 6% (19:1) pre-mixed acrylamide-bisacrylamide and 1× TBE. The gel was pre-run until reaching ~50˚C, and the sample was resolved on gel by running at 25 W for 2–3 h at room temperature. The RNA band was visualised by UV shadowing, the correct band excised, and eluted by crushing the gel and incubating in ddH$_2$O with shaking overnight at 4˚C. The resulting suspension was filtered to remove the gel pieces and exchanged into MWQ and concentrated using ethanol precipitation. The RNA was stored at −80˚C until use.

## Cy5-labelling of RNA

4.5S and RNA$^{s4.5S}$ were labelled using 5' EndTag™ DNA/RNA Labeling Kit (Vector Laboratory, MB-9001) and Cy5-maleimide (Kerafast lnc., MA, USA) by following manufacturer's protocol. Briefly, 0.6 nmol of RNA was mixed with alkaline phosphatase in universal reaction buffer (supplemented in the kit) and incubated for 30 min at 37˚C. T4 polynucleotide kinase and ATPγS (supplemented in the kit) were then added to the mixture, followed by 30 min incubation at 37˚C. The mixture was then reacted with Cy5-maleimide dissolved in DMSO for 2 h at room temperature in the dark. The RNA was purified by phenol-chloroform extraction and concentrated by ethanol precipitation. The labelled RNA was redissolved in water and stored at −80˚C until use.

## Preparation and refolding of PNA$^{tet}$ and variants, RNA$^{tet}$, moRNA$^{tet}$ and 4.5S RNA

PNA$^{tet}$ (NH$_2$-GUCC G^GAA GGAC-AEEA-CONH$_2$), PNA$^{tetS}$ (NH$_2$-GCC G^GAA GGC-AEEA-CONH$_2$), PNA$^{tetL}$ (NH$_2$-GAUCC G^GAA GGAUC-AEEA-CONH$_2$) and PNA-$^{tetN}$ (NH$_2$-GAUCC T^UCG GGAUC-AEEA-CONH$_2$) were ordered from PANAGENE (South Chungcheong, South Korea). G^ is Glu-gamma-Guanine, T^ is Glu-gamma-Thymine and AEEA is a2-aminoethoxy-2-ethoxy acetic acid linker. To enhance the stem stability, a GC pair is placed at the terminal ends. For PNA$^{tetL}$, the addition intervening base-pairs were chosen to be AU and UA pairs to aid with potential Nuclear Magnetic Resonance (NMR) spectral interpretation and assignment and lower the GC content. Biotinylated PNA$^{tet}$ with an N-terminal biotin group as well as PNA$^{tet}$ without the AEEA-linker (NH$_2$-GUCC G^GAA GGAC--CONH$_2$) were also ordered from PENAGENE. RNA$^{tet}$ and moRNA$^{tet}$ (5′- rG*rC*rG rCrCrG rGrArA rGrGrC* rG*rC -3′, where rN* = modified nucleotide with phosphorothioate backbone) and IN3E3_LSA (20-base RNA with a 5-base-paired stem, phosphorothioate backbone and 2′-OMe-modified nucleosides [27]) was ordered from IDT (Singapore). The lyophilised PNAs and RNA oligos were solubilised in nuclease-free water to 100 μM and 750 μM, respectively and stored at -80˚C. Before using, the PNA or RNA were freeze-dried and reconstituted in a buffer containing 20 or 50 mM HEPES, pH 7.5, 150 mM NaCl and 3 mM MgCl$_2$. The PNA and RNA samples were then refolded by two cycles of incubation at 95˚C for 1 min followed by 10 min on ice.

4.5S RNA and RNA$^{s4.5S}$ were stored in nuclease-free water at -80˚C. For refolding, HEPES, pH 7.5 was added to 50 mM, followed by heat treatment in the same manner.

## Nuclease and proteinase degradation assays

In a 10-μL reaction, PNA$^{tet}$ (100 μM) or RNA$^{tet}$ (80 μM) was incubated with 0.1 mg/mL each of DNase I (Sigma-Aldrich, Cat. No.: D7291), RNase A (ThermoFisher, Cat. No.: EN0531) and proteinase K (NEB, Cat. No.: P8107S) in the refolding buffer for 60 min at 37˚C. Control samples containing the same amount of PNA$^{tet}$ and RNA$^{tet}$ was treated in the same manner as per legend.

Following incubation, samples for reverse-phased High Performance Liquid Chromatography (rpHPLC) were diluted 100-fold into Buffer A (100% MQW, 0.1% trifluoroacetic acid), centrifuged at 15,000 rpm for 5 mins and the supernatant injected into an Agilent 1260 series HPLC system fitted with an Eclipse-XDB C18 5 μm × 230 mm reverse-phase HPLC column (Agilent). A linear gradient from 5–90% Buffer B (100% acetonitrile, 0.1% trifluoroacetic acid) in Buffer A over 20 min was used to elute the sample components. The total run time including equilibration and ramping back to 5% Buffer B was 30 min with a flow rate of 1 mL/min. UV detector wavelengths were set to 215, 260 and 280 nm. Note the rpHPLC trace for PNA[tet] alone terminated at ~20 min due to an instrument error.

For RNA[tet] and moRNA[tet] samples used for RNA gel electrophoresis, 5 μM RNA oligo was treated with each enzyme separately (0.3 mg/mL DNase I, 0.3 mg/mL RNase A, or 2 mg/mL proteinase K) for 60 min at 37˚C. The reaction products were mixed with 2× RNA loading dye and separated on a 16% denaturing TBE gel by electrophoresis at 300 V for 25 min at room temperature. The gel was imaged by SYBR™ Gold stain.

## Recombinant expression of unlabelled FtsY$_{NG}$ and Ffh

The NG domain of FtsY (residues 196–498, referred to as FtsY$_{NG}$) and C-terminal truncation of Ffh (residues 1–432) containing an N-terminal His$_6$-tag were inserted between the NcoI and BamHI sites of pET15b vectors as described in [28]. The plasmids were transformed into BL21 Rosetta cells and the freshly transformed cells were grown in 1 L LB medium supplemented with 100 μg/mL ampicillin and 34 μg/mL chloramphenicol at 37˚C until the absorbance at 600 nm reached 0.6. Protein expression was induced with 0.5 mM IPTG, and the cells were grown at 28˚C for an additional 3.5 h, before harvested by centrifugation (5,000×g for 20 min at 4˚C).

## Recombinant expression of $^2H^{13}C^{15}N$-labelled FtsY$_{NG}$

Isotopically labelled FtsY$_{NG}$, used for $^{15}N$-$^1H$-TROSY-HSQC titrations and assignments, was expressed as previously described [29] with 10 g/L $^{13}C$ glucose as the sole carbon source and 5.2 g/L $^{15}N$ ammonium chloride as the sole nitrogen source in 100% $^2H_2O$. Protein expression was induced with 1 mM IPTG at an optical density (OD600) of 3, and cells were harvested after exhaustion of the carbon source, as indicated by a small rise in pH, at an OD600 of 10.2. The non-exchangeable deuteration level of the $^2H^{13}C^{15}N$-labelled FtsY$_{NG}$ was determined, by partial trypsin digest MALDI-TOF, to be 87%.

## Purification of FtsY$_{NG}$, Ffh and Nsp9

Ffh pellet from 1L of cell culture was resuspended in 30 mL of lysis buffer (50 mM HEPES pH 7.5, 300 mM NaCl, 10 mM MgCl$_2$, 0.1% Triton-X 100, 5 mM imidazole, 1 mM DTT, 1 mM PMSF and 1× cOmplete™ Protease Inhibitor Cocktail, Roche), and lysed by sonication. The lysate was clarified by centrifugation at 15,000×g for 30 min at 4˚C and the soluble fraction was mixed with 5 mL Ni-NTA Agarose resin (Invitrogen, Cat. No.: R901-15) pre-equilibrated in Ffh lysis buffer. After incubating for 1 h on a shaker at 4˚C, the beads were washed 3 times with 5× CV of wash buffer 1 (50 mM HEPES pH 7.5, 300 mM NaCl, 5% glycerol, 10 mM MgCl2, 10 mM imidazole, 1 mM TCEP) and then 3 times with wash buffer 2 (same as wash buffer 1 but with 20 mM imidazole). The protein was eluted with elution buffer (same as wash buffer 1 but with 400 mM imidazole). The eluted protein was pooled and mixed with TEV protease, followed by dialysis against 20 mM HEPES pH 7.5, 300 mM NaCl, 5% glycerol, 1 mM TCEP overnight at 4˚C. The cleaved protein was separated from the uncleaved and TEV protease by reverse Ni-NTA chromatography and buffer exchange into 20 mM HEPES pH 7.5, 100

mM NaCl, 5% glycerol, 1 mM TCEP with dialysis. Ffh was further purified with cation exchange chromatography using a HiTrap™ SP HP column (Cytvia), with a gradient across 0 −1 M NaCl over 20 CV. The eluted peak containing Ffh was concentrated and injected onto a HiLoad 16/600 Superdex 75 pg SEC column (Cytiva) pre-equilibrated in RNase Free SEC buffer (50 mM HEPES pH 7.4, 10 mM $MgCl_2$, 300 mM NaCl, 10% glycerol, 1 mM TCEP). The final purified protein was concentrated and stored at -80˚C.

FtsY$_{NG}$ was purified in a similar manner. Except the protein eluted from Ni-NTA affinity chromatography was dialysed against 50 mM MES pH 6, 100 mM NaCl, 1 mM TCEP, 5% glycerol without TEV protease. On the next day, the protein was loaded onto a HiTrap™ SP HP column, and cation exchange chromatography was performed using 50 mM MES pH 6, 1 mM TCEP, 5% glycerol with and without 1 M NaCl. The later peak eluted from cation exchange column with a lower A260:280 ratio was mixed with TEV protease and digested while dialysing against 100 mM HEPES, pH 7.4, 200 mM NaCl, 5% glycerol, 1 mM TCEP for 48 h at 4˚C. The cleaved FtsY$_{NG}$ was separated using two rounds of reverse Ni-NTA chromatography and injected onto Superdex 75 column using the same SEC buffer. The purification protocols are the same for triple labelled and unlabelled FtsY$_{NG}$. NMR buffer used for FtsY$_{NG}$ HSQC titration experiments was 50 mM sodium phosphate, 3 mM $MgCl_2$, 150 mM NaCl, 1 mM DTT.

Nsp9 was expressed and purified as described in [30]. Briefly, $^{15}N$-labelled 6xHis-Nsp9 from SARS-CoV-2 was expressed in *Escherichia coli* BL21(DE3) cells in a bio-fermenter using the protocol described in [31]. Cells were lysed by sonication in lysis buffer (20 mM Tris pH 8.0, 50 mM NaCl, 3 mM TCEP, 0.5 mM PMSF, 0.1% Triton X-100), were centrifuged, and the supernatant was subjected to Ni-NTA affinity chromatography followed by thrombin cleavage for 1 h at 25˚C and then 15 h at 4˚C to remove the His$_6$-tag. The eluate from Ni-NTA affinity chromatography was subjected to size exclusion chromatography using a Superdex 75 column (120 mL) equilibrated with NMR buffer (25 mM sodium phosphate pH 6.0, 150 mM NaCl, 1 mM DTT).

## Circular dichroism (CD) spectroscopy

CD samples were prepared by refolding RNA$^{tet}$ and moRNA$^{tet}$ at 30 μM in 10 mM HEPES, pH 7.0 and 0.5 mM $MgCl_2$. Experiments were conducted using a JASCO J-815 CD Spectrometer with 0.1-cm pathlength quartz cuvettes. CD spectra at 20˚C and 90˚C were recorded over 200–300 nm at a rate of 50 nm/min with a bandwidth of 1 nm and a D.I.T of 1 sec. Spectra shown were averaged over three scans. Melt curves from 20–90˚C was recorded at 265 nm and 210 nm with a ramp of 1˚C/min, collected in 2˚C intervals with a bandwidth of 1 nm and D.I. T of 4 sec.

## NMR spectroscopy

All NMR spectra were recorded on Bruker Avance III 600 or 800 MHz Spectrometers equipped with a TCI cryogenic probehead. Acquisition temperatures ranged from 2–25˚C and are stated in the methods, results and/or legends. All NMR samples were loaded into 3- or 5-mm NMR tubes with 0.5–1.5 mM DSS and 5% (v/v) $D_2O$ added to each sample prior to acquisition.

NMR samples were prepared as described in the respective sections. The PNA$^{tet}$ presented in **S1 Fig in S1 File** was the version without the AEEA-linker. For NMR experiments, purified FtsY$_{NG}$ and Nsp9 proteins were dialysed in NMR buffers for at least 2 h at 4˚C using a 10-kDa MWCO dialysis membrane. PNA/RNA was added to isotopically labelled FtsY$_{NG}$/Nsp9 at 0.5 to 2 molar ratio as indicated and NMR spectra collected at 25˚C before and after the PNA/

RNA addition. As Nsp9 gave high quality NMR spectra at concentrations as low as 40 μM, the PNA[tet] used in the titration did not contain the AEEA-linker. For FtsY$_{NG}$ titrations, sample concentrations ranged from 80–120 μM. 1D $^1$H spectra were acquired using the standard zgesgp program (SW = 20 ppm) from Bruker pulse library to suppress the water peak. 2D $^{15}$N-$^1$H TROSY-HSQC (Transverse Relaxation Optimised Spectroscopy Heteronuclear Single-Quantum Coherence) spectra were acquired using b_trosyetf3gpsi.3 (SW = 12 and 26 ppm, for $^1$H- and $^{15}$N-dimensions, respectively) from the Bruker pulse library. Relaxation delay for the 1D $^1$H and 2D $^{15}$N-$^1$H TROSY-HSQC was set to 1 s and 0.3 s, respectively. The spectra were processed and analysed using Topspin 3.6.x or 4.1.x (Bruker, Biospin) with the $^1$H chemical shift of the DSS trimethylsilyl group used as the reference. Number of scans (NS) were adjusted depending on concentration of the proteins after dialysis to obtain sufficient signal intensity and ranged from 64–256 scans.

1D $^1$H spectra were collected after each HSQC experiment as quality control experiment to ensure the protein/PNA[tet]/RNA[tet]/moRNA[tet] have not been degraded during acquisition of the 2D $^{15}$N-$^1$H TROSY-HSQC spectra.

For assignment of FtsY$_{NG}$ in 2D $^{15}$N-$^1$H TROSY-HSQC spectra, 3D TROSY HNCA, HN(CO)CA, HNCB, HNCACB, HN(CO)CACB spectra were collected on an 800 MHz spectrometer at 25˚C from multiple $^2$H$^{13}$C$^{15}$N-FtsY$_{NG}$ samples ranging from 150–300 μM in FtsY$_{NG}$ NMR buffer for manual assignments by CARA (http://cara.nmr.ch/doku.php). The assigned chemical shifts were deposited in BMRB (ID 52588). The backbone assignments of the RNA binding site were further validated when they corresponded to amide chemical shift changes in the various titration experiments.

For assignment of imino and imide signals for PNA[tetL], 2D $^1$H-$^1$H NOESY spectra (Bruker sequence noesyesgpph) were collected at 10˚C with 4096 and 400 points in the time domain for F2 and F1, respectively, with a NOE mixing time of 100–200 ms and NS of 96 scans. NMR data were processed using Topspin (Bruker) and analysed with SPARKY (T. D. Goddard and D. G. Kneller, University of California at San Francisco).

## Surface plasmon resonance (SPR)

All measurements were collected on a Biacore T200 (Cytiva), and all data analysed using the Biacore Insight Evaluation Software. Biotinylated PNA[tet] was immobilised to an SA chip (Cytiva) at 10˚C in 20 mM HEPES, 150 mM NaCl, 3 mM MgCl$_2$ to levels of 510 RU. FtsY$_{NG}$ (1.6–100 μM) was titrated onto immobilised PNA[tet] at 10˚C in 20 mM HEPES, 150 mM NaCl, 3 mM MgCl$_2$, 1 mM TCEP, 1% (v/v) glycerol, 0.005% (v/v) Tween pH 7.5.

Avi-FtsY$_{NG}$ or Avi-Nsp9 were coupled to a streptavidin surface, which was prepared by amine coupling streptavidin to a CM5 chip (Cytiva) using standard methods. Briefly, the surface was activated with a 1:1 mixture of 1-ethyl-3-(3-dimethylaminopropyl) carbodiimide hydrochloride (0.4 M) and *N*-hydroxysuccinimide (0.1 M). Streptavidin (2 μg/ml) in 10 mM sodium acetate (pH 5) was injected for 450 s at 2 μl/min. The surface was then blocked with an injection of 1 M ethanolamine (pH 8.5). Avi-FtsY$_{NG}$ was immobilised at 25˚C in 20 mM HEPES, 150 mM NaCl, 3 mM MgCl$_2$ pH 7.5 to levels of 4650 RU, and Avi-Nsp9 to levels of 520 RU PNA[tet] (0.28–17.5 μM) and moRNA[tet] (0.8–50 μM), along with GDP (0.1–375 μM) and IN3E3_LSA (0.1–50 μM) as positive controls were titrated over immobilised Avi-FtsY$_{NG}$ and Avi-Nsp9 at 10˚C, in 20 mM HEPES pH 7.5, 150 mM NaCl, 3 mM MgCl$_2$, 1 mM TCEP, 1% glycerol, 0.005% Tween. Note higher concentration points (> 20 μM) for titration of PNA[tet] into Avi-FtsY$_{NG}$ were removed from the fitting as the sensorgram shape was poor, possibly due to PNA[tet] aggregation and/or precipitation upon interaction with Avi-FtsY$_{NG}$ on the chip surface.

### Electrophoretic mobility shift assay (EMSA)

Cy5-labelled RNA, PNA$^{tet}$ and moRNA were prepared as described above. Each reaction contained 20 nM of Cy5-labelled RNA, 50 mM HEPES, pH 7.5, 150 mM KCl, 1.5 mM MgCl$_2$, 10% glycerol, 0.01% (v/v) Igepal and 1 mM GMP-PNP. Ffh was added to 200 nM first and incubated for 10 min at room temperature. FtsY$_{NG}$ was pre-mixed and incubated with 20 μM of refolded PNA$^{tet}$ separately at room temperature for 30 min, before adding to the Ffh-RNA mixture to final concentration of 400 nM. The samples were loaded onto a 1× TBE 6% polyacrylamide gel containing 2.5 mM MgCl$_2$ after exaction. Following a 5-min incubation, and the gel was run in 0.5× TBE buffer containing 2.5 mM MgCl$_2$ at 1 W/10 mL gel for 2 h at 4˚C. The gel was then imaged on an FLA-9000 laser scanner.

### Microscale thermophoresis (MST)

Cy5-labelled RNA$^{s4.5S}$ and PNA$^{tet}$ and variants were prepared and refolded as described above. FtsY$_{NG}$ was buffer exchanged into MST buffer (50 mM HEPES, pH 7.5, 100 mM NaCl, 1.5 mM MgCl$_2$, 10% glycerol) and concentrated to ~750 μM using Amicon Ultra-0.5 mL Centrifugal Filters (10 K MWCO, Merck Millipore). For each replicate, a set of titration points FtsY$_{NG}$ was prepared by 1:2 serial dilution using the same MST buffer, then mixed with 50 nM Cy5-labelled RNA$^{s4.5S}$ and 0.01% (v/v) Igepal, with or without 5 μM PNA$^{tet}$ and variants. Following 10 min incubation at room temperature, the samples were loaded into standard capillary tubes (Nanotemper) before undergoing MST in a Monolith™ NT.115 instrument. Thermophoresis was conducted at 90% LED power, and 20% MST power. Thermophoresis data were then analysed with MO Affinity Analysis software v2.3.

## Results

### Design of PNA constructs to mimic the 4.5S tetraloop

RNA hairpins are a common folded structure consisting of a Watson-Crick base-paired stem structure and an unpaired loop sequence. Hairpins are typically found in transcription termination and for displaying certain bases for sequence-specific interaction with RBPs. The 4.5S RNA has a tetraloop displaying a GNRA (GGAA) sequence followed by a base-paired stem region and the tetraloop is responsible for the initial interaction with FtsY. Furthermore, the crystal structure of 4.5S RNA tetraloop binding to the FtsY:Ffh NG heterodimer indicates that a short stem of the hairpin structure is sufficient for the initial assembly of the complex [32]. As the 4.5S RNA recognition site mainly covers the GGAA tetraloop and the two most adjacent CC-GG base-paired stem, the designed PNA constructs contain this core sequence with an additional one to three base pairs (termed PNA$^{tetS}$, PNA$^{tet}$ and PNA$^{tetL}$, respectively; **Fig 1A**) at the end to promote the stability of the stem. We also tested PNA$^{tetN}$ which has the same stem as PNA$^{tet}$ and a loop with a non-binding sequence UUCG (in our PNA is TUCG) [33]. As positive controls, RNA$^{tet}$ and moRNA$^{tet}$ (modified RNA$^{tet}$) with a five base pairs stem were used since this is the minimum length in our experience to achieve reliable hairpin formation and FtsY binding. A major motivation for using PNAs as RNA mimics for RBP binding is the increased stability under a range of conditions, including a non-RNase-free setup. Therefore, moRNA$^{tet}$, which has the first and last two phosphodiester bonds replaced by phosphorothioate bonds (**Fig 1B**) with increased degradation resistance over RNA$^{tet}$, is used as a comparison to PNAs. RNA$^{tet}$ and moRNA$^{tet}$ share nearly identical circular dichroism (CD) spectra as well as thermal melt profiles indicating similar structures and thermal stability (**S1 Fig in S1 File**).

Because PNA base-pairing is thought to be more robust than in RNA, fewer than five base-pairs in the stem might be sufficient for hairpin formation. In addition, all PNA$^{tet}$ variants

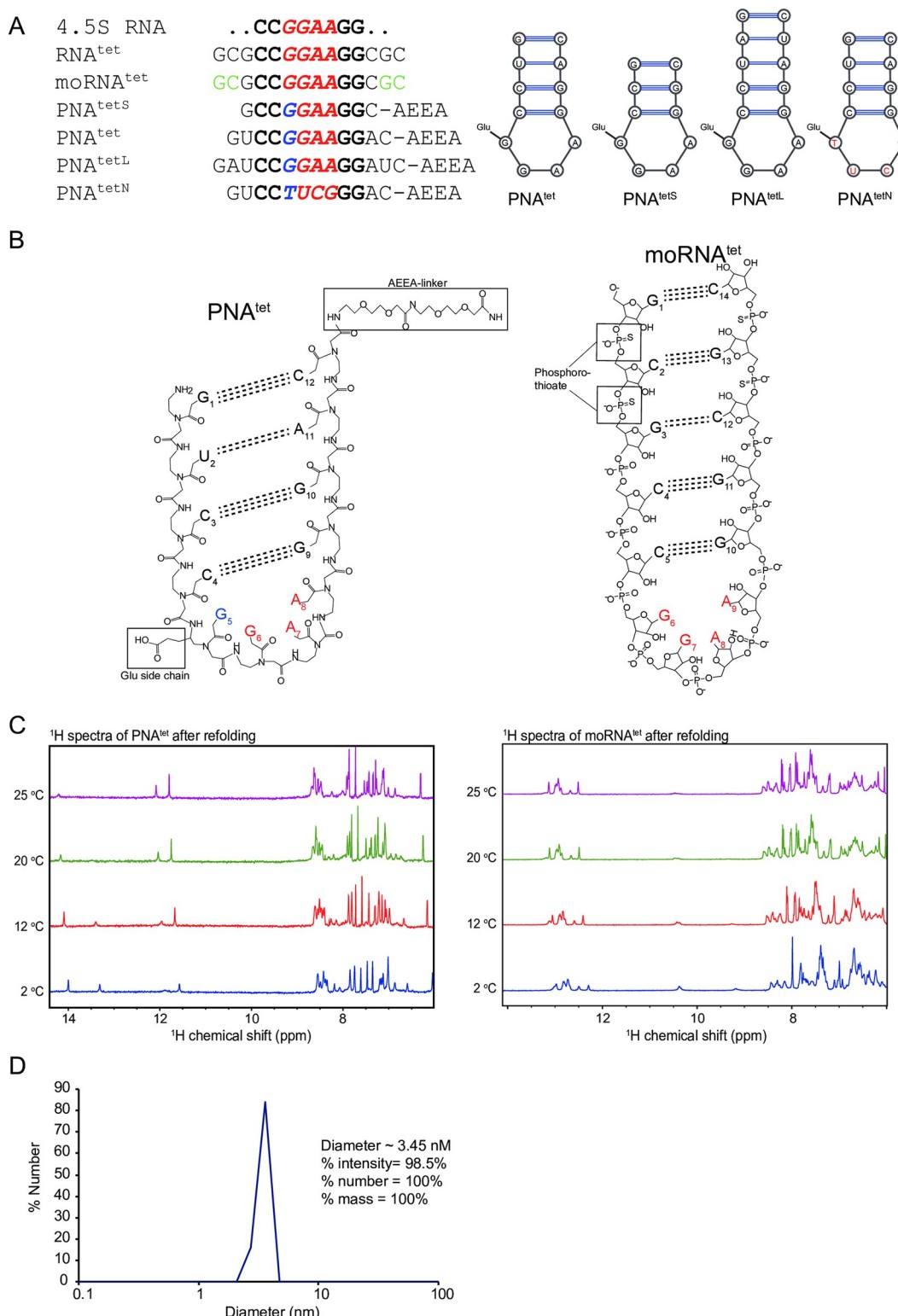

**Fig 1. Structure and stability assessments of folded PNA<sup>tet</sup> and moRNA<sup>tet</sup>.** (A) Sequence (*left*) and schematic (*right*) comparison of PNA and RNA molecules used in this study. The core FtsY-binding region in 4.5S RNA, consisting of the GGAA tetraloop (red italic) and the two base pairs adjacent are highlighted in bold. Bases shown in green in moRNA<sup>tet</sup> indicate phosphotioester backbone modification. Blue indicates glutamic acid side chain on the gamma carbon of the first G. (B) Predicted folded structure of PNA<sup>tet</sup> and moRNA<sup>tet</sup> and their backbone modifications. (C) 1D <sup>1</sup>H NMR spectra of

100 μM folded PNA^tet and moRNA^tet at various temperatures. (D) Particle distribution of 50 μM folded PNA^tet in 50 mM HEPES pH 7.5, 150 mM NaCl, 3 mM MgCl$_2$, measured by DLS at 25˚C.

contain a glutamic acid side chain on the gamma carbon of the first G in GGAA or the first T in TUGG (since the modification available on Gs and Us only) (Fig 1B) to mimic the electrostatic interactions between the negatively charged backbone in 4.5S RNA and the basic residue patch (K399, R402, K405 and K406) on FtsY [32]. We also included AEEA-linkers at the C-terminal of the PNAs to enhance solubility (Fig 1B).

## PNAs can fold into a stable hairpin structure

Most of the reported PNA structures and applications thus far correspond to PNA-PNA or PNA-NA duplexes/triplexes [15, 16], even though the ability of PNAs to form stable hairpins was demonstrated in the 1990s [34]. Typically, PNAs adopt a P-form helix in a PNA-PNA or PNA-NA base pairing, which is more conformationally restrictive than standard NA backbones adopting the more common B-helix [35]. Therefore, we first tested if the designed PNAs could fold into a hairpin and sufficiently mimic the 4.5S RNA stem-loop for FtsY binding. When Watson-Crick base-pairing occurs, the imino protons of Guanosine and Uracil bases are in hydrogen bonding with Cytosine and Adenine bases, respectively, and are protected from exchange with water. As such, only protected hydrogen-bonded imino protons can be detected as peaks between 10 and 15 ppm in one-dimensional (1D) $^1$H Nuclear Magnetic Resonance (NMR) spectra [36]. As other hydrogen atoms in nucleic acids do not feature in this region, the presence of signals between 10–15 ppm (imino region) can be used to indicate the formation of Watson-Crick base-pairing. In addition, the position of NMR peaks on the chemical shift scale is sensitive to the local chemical environment experienced by the hydrogen atoms. Therefore, an imino proton participating in a stable hydrogen bond in a well-folded duplex structure would be expected to give rise to one sharp peak, and the number of imino peaks can be used to indicate the number of Watson-Crick base pairs present [37].

When PNA^tet was initially resuspended in water, the NMR spectrum showed no clear peaks in the imino region (S2 Fig in S1 File) indicating the absence of stable base-pairing. After refolding PNA^tet, PNA^tetS, PNA^tetL, PNA^tetN and moRNA^tet using heat-cooling cycles in refolding buffer (20 mM HEPES, pH 7.5, 150 mM NaCl, 3 mM MgCl$_2$), 1D $^1$H NMR spectra of all samples display clear imino signals (Fig 1C and S3 Fig in S1 File). Compared with the PNA^tet spectra, the imino peaks in the moRNA^tet spectra are broader (Fig 1C), suggesting a range of local chemical environments are experienced by the imino hydrogens and conformational exchange in moRNA^tet. The overlapping imino peaks of moRNA^tet are clustered at 12–13.2 ppm, consistent with published imino chemical shifts from other short dsRNA where the imino peaks of G and U generally occurring in the ~10–13 ppm and ~13–14 ppm regions, respectively [38, 39]. In contrast, sharp and discrete imino signals, albeit at different intensities, can be observed in the PNA^tet, PNA^tetS, PNA^tetL and PNA^tetN spectra (S3 Fig in S1 File), indicating that the PNAs have folded into a stem loop with base pairing observed for all bases in the stem. At 25˚C, generally one to two fewer imino peaks were observed consistent with the expectation that the two ends of the hairpin are likely to experience some opening and closing in solution at the higher temperatures (Fig 1C and S3 Fig in S1 File). From the NMR results, large-scale synthesis of PNA^tet and PNA^tetL was ordered for further investigations as these PNAs display at least three well-protected imino groups in the stem region at 2–25˚C.

We attempted to prepare PNA^tet and PNA^tetL samples at >200 μM to collect 2D $^1$H-$^1$H NOESY spectra. While PNA^tet generally appeared more soluble than PNA^tetL and was able to dissolve at ~220 μM, precipitation was apparent during NOESY collection and this impacted

spectral quality. PNA$^{tetL}$ was only able to be dissolved at ~150 μM and some precipitation was also observed during NOESY acquisition. However, the NOESY spectra of PNA$^{tetL}$ were of better quality than for PNA$^{tet}$ and were therefore used for sequence-specific partial assignments of the imino and imide signals (**S4A Fig in S1 File**). A number of NOEs from hydrogens in the base-paired nucleobases can be detected (*e.g.*, G11H2-U3H3, A2H6-U13H3, C4H5-G11H1, C14H5-G1H1, see **S4B Fig in S1 File** for a schematic of the proposed basepairing). The linewidths of the imino groups also agree well with their positions in the stem loop, with the middle bases (U13, U3, G11) displaying the sharpest imino signals, while the G10 (adjacent to the tetraloop bulge) displaying a broad imino signal (**S4C Fig in S1 File**).

Since NMR spectra of all PNAs displayed sharp and discrete signals corresponding to the expected number of base pairs at least at lower temperatures and PNA duplexes have previously been shown to have much lower mismatch tolerance than RNA and DNA [40], the possibility of PNAs to form alternative base pairing is considered low. Subsequent experiments requiring a high PNA concentration were conducted on PNA$^{tet}$ as it is generally more soluble than PNA$^{tetL}$. To determine whether the observed base-pairing was intra- or inter-molecular, dynamic light scattering (DLS) measurements were conducted on PNA$^{tet}$ (**Fig 1D**) to identify the population of species in solution after refolding. Clearly, one dominant species (98.5% by intensity, 100% by the number of molecules) is present after refolding. As intermolecular base pairing of PNA$^{tet}$ will cause unpaired bases to interact with other PNA molecules and lead to oligomerization, the DLS result supports that PNA$^{tet}$ exists as a monomer after refolding, consistent with the sharp signals observed in the imino region. The estimated hydrodynamic radius of the dominant species from DLS is ~17 Å, also agreeing with an intramolecular hairpin stem loop formed from 12 bases.

## PNA$^{tet}$ and PNA$^{tetL}$ inhibit FtsY$_{NG}$ interaction with 4.5S RNA and disrupt SRP:FtsY$_{NG}$ complex formation

Given PNA$^{tet}$ and variants can form stable hairpins, we tested if they can sufficiently mimic the 4.5S tetraloop and compete for FtsY$_{NG}$ binding in *in vitro* assays including MicroScale Thermophoresis (MST) and EMSA.

MST competition assays were adopted to evaluate the inhibition activity of PNA$^{tet}$ and variants. In this assay, a short 41-nt RNA displaying the GGAA tetraloop, described in [32], and carrying a 5′ Cy5-tag, named shortened 4.5S RNA (RNA$^{s4.5S}$), was used. The affinity between FtsY$_{NG}$ and RNA$^{s4.5S}$ was measured in the absence and presence of PNA$^{tet}$ or variants at 5 μM in triplicates (**Fig 2A**) and the fold change in K$_D$ calculated (**Fig 2B**). PNA$^{tet}$ and PNA$^{tetL}$ increased the K$_D$ between RNA$^{s4.5S}$ and FtsY substantially suggesting that PNA$^{tet}$ and PNA$^{tetL}$, but not PNA$^{tetS}$ and PNA$^{tetN}$, can compete with RNA$^{s4.5S}$ for binding to FtsY$_{NG}$.

Next, we tested the ability of PNA$^{tet}$ to prevent the complex formation between FtsY and the SRP by competing with 4.5SRNA tetraloop for the open complex formation. The protein component of SRP, Ffh, binds tightly to 4.5S RNA via its M-domain and together they bind to FtsY. Upon the initial FtsY:4.5S tetraloop interaction, also known as open complex, the Ffh:FtsY NG domain heterodimer complex forms in the presence of GTP to generate the SRP:FtsY complex. In this experiment, FtsY$_{NG}$ was pre-incubated with or without 20 μM of PNA before adding SRP and GMPPNP. GMPPNP, a non-hydrolysable GTP analogue, was used to lock in the formation of a Ffh:FtsY$_{NG}$ heterodimer, a GTP-mediated process, so the complex could be detected using EMSA [41, 42]. As shown in **Fig 2C**, we observed that 4.5S RNA was fully shifted by Ffh protein, which was then super shifted by the addition of FtsY$_{NG}$. However, in the presence of PNA$^{tet}$, the amount of ternary complex formation was reduced, indicating that PNA$^{tet}$ disrupted the binding of SRP to its receptor. Similarly, moRNA$^{tet}$ also displayed a

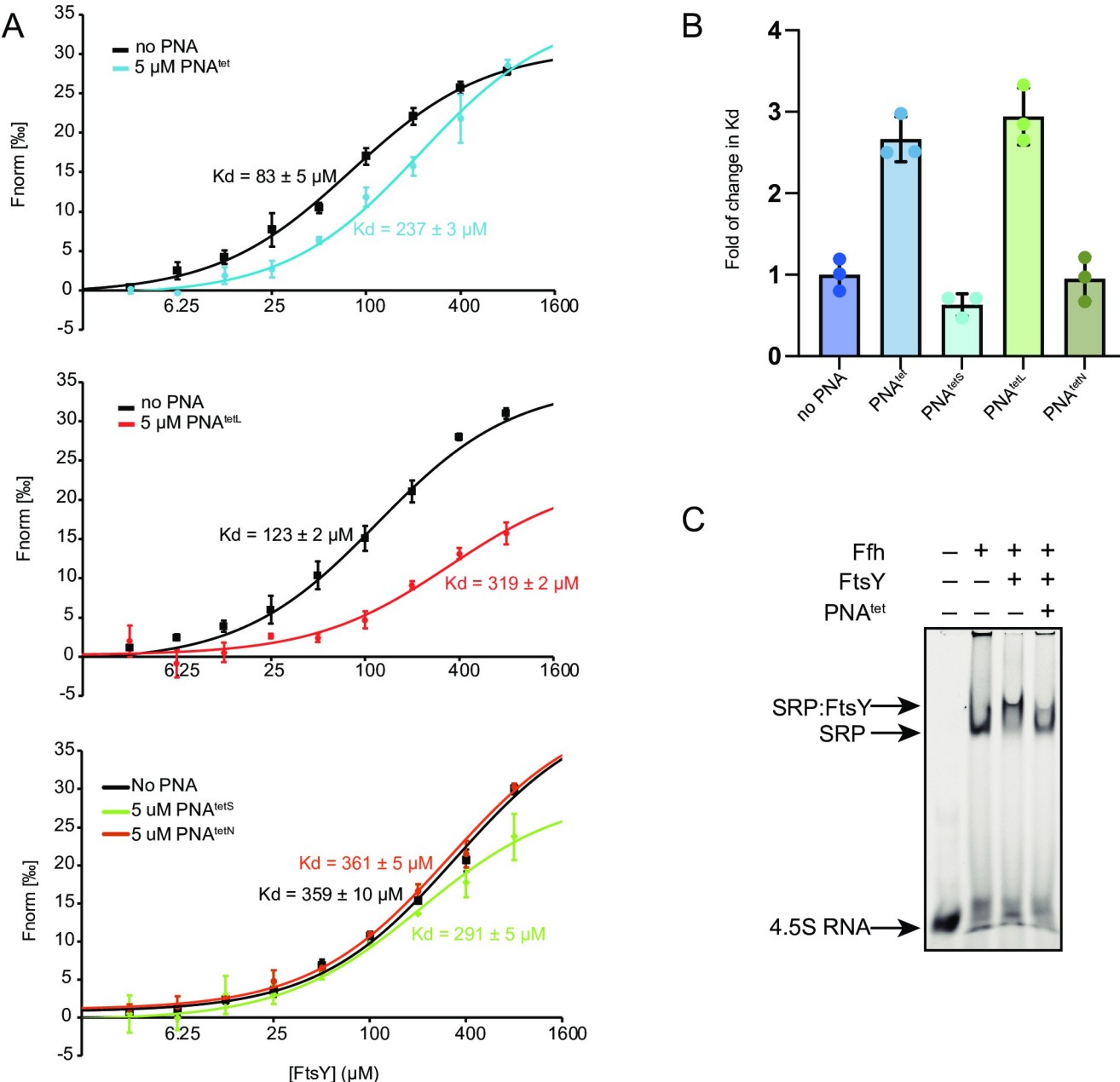

**Fig 2. PNAᵗᵉᵗ showed inhibitory effect on SRP:FtsYₙG assembly.** (A) MST showing that folded PNAᵗᵉᵗ and PNAᵗᵉᵗᴸ reduces the affinity between FtsYₙG and RNAˢ⁴·⁵ˢ. Cy5-labelled RNAˢ⁴·⁵ˢ was titrated with FtsYₙG with or without the presence of 5 μM folded PNAᵗᵉᵗ or PNAᵗᵉᵗᴸ. Three technical replicates (error bar = Stdev) and a fitting curve to a 1:1 binding isotherm are shown, and $K_D$ values are indicated. (B) The effect of folded PNAᵗᵉᵗ and variants on $K_D$ between FtsYₙG and RNAˢ⁴·⁵ˢ. Three technical replicates (error bar = SD) of each experiments were shown, the apparent $K_D$ of each PNA is normalised against $K_D$ without PNA. (C) An EMSA gel showing PNAᵗᵉᵗ inhibiting interaction between SRP and FtsYₙG. Each lane contains 20 nM of Cy5-labelled RNA⁴·⁵ˢ and 1 mM GMP-PNP. Ffh was incubated with RNA for 10 min at room temperature, before mixing with 400 nM FtsYₙG with or without pre-incubating with 20 μM PNAᵗᵉᵗ.

similar level of inhibition on SRP:FtsYₙG complex formation, and the inhibition by PNAᵗᵉᵗ/moRNAᵗᵉᵗ was dose dependent (**S5 Fig in S1 File**). It is worth noting that in the presence of GMPPNP, the complex formation is biased towards the accumulation of SRP:FtsYₙG over time and PNAᵗᵉᵗ/moRNAᵗᵉᵗ only acts to slow down SRP:FtsYₙG formation. Therefore, their

inhibitory activities were not quantified in this assay. Overall, the *in vitro* assays supported that PNA$^{tet}$ can inhibit the assembly of SRP:FtsY$_{NG}$ complex by competing with the binding of the GGAA tetraloop in 4.5S RNA to FtsY$_{NG}$.

## Folded PNA$^{tet}$ and moRNA$^{tet}$ bind to the RNA-binding face of FtsY$_{NG}$

One major advantage of using PNAs over RNAs in RBP interaction studies is PNAs are completely resistant to RNase, DNase and proteinase K attacks. We have confirmed that PNA$^{tet}$ maintained A260 peak intensity in reverse-phase high pressure liquid chromatography (rpHPLC) spectra after one hour of enzyme challenge (**S6A Fig in S1 File**) while the A$_{260}$ peak from RNA$^{tet}$ decreased significantly. As expected, moRNA$^{tet}$ was also more resistant to degradation than RNA$^{tet}$ according to RNA gel electrophoresis (**S6B Fig in S1 File**) and hence was used in subsequent assays to compare with PNA$^{tet}$.

The mechanism by which PNA$^{tet}$ interferes with SRP:FtsY$_{NG}$ formation was assessed using surface plasmon resonance (SPR) and NMR experiments. In SPR assays, RNase-free conditions are particularly hard to establish but no longer a concern for titrating PNA$^{tet}$ and moRNA$^{tet}$ over immobilised FtsY$_{NG}$. Binding of moRNA$^{tet}$ and PNA$^{tet}$ to FtsY$_{NG}$ was observed (**Fig 3A** and **3C**), however, for both molecules the binding curves were not complete over the assayed concentration range. Using GDP binding as a control to allow fix fitting [19], for moRNA$^{tet}$ an estimated dissociation constant (K$_D$) of 160 ± 20 µM (average ± stdev from three replicate experiments, **Fig 3B**) and for PNA$^{tet}$ an estimated K$_D$ of 440 ± 30 µM were calculated (fit and error for one independent experiment, **Fig 3D**). When the orientation of the experiment was reversed (*i.e.*, biotinylated PNA$^{tet}$ was immobilised and FtsY$_{NG}$ added in increasing concentrations), dose-dependent responses were also observed with a linear response over the assayable concentration range (**Fig 3E**).

To assess whether PNA$^{tet}$ binds to FtsY$_{NG}$ in the targeted RNA-binding site, Transverse Relaxation-Optimised Spectroscopy Heteronuclear Single Quantum Coherence (TROSY-HSQC) NMR experiments were performed similar to those previously conducted for fragment screening against FtsY$_{NG}$ [19]. As shown in **Fig 3F**, the addition of the PNA$^{tet}$ into isotopically labelled $^2$H$^{13}$C$^{15}$N-FtsY$_{NG}$ at 0.5 to 2 molar equivalence resulted in a subset of the peaks losing intensity or moving in the $^{15}$N-$^1$H-TROSY-HSQC spectra indicative of perturbations of specific residues. Encouragingly, the addition of moRNA$^{tet}$ produced similar spectral changes (**Fig 3G**) indicating the same binding mode and similar binding residues for both molecules. While the perturbations are smaller for PNA$^{tet}$ than for moRNA$^{tet}$ titrations consistent with the affinity difference as measured by SPR, the smaller changes are also likely contributed by the limited solubility of PNA$^{tet}$ and a fine precipitation was observed at later titration points for PNA$^{tet}$. We note that the peaks that experienced the largest intensity changes upon PNA$^{tet}$ and moRNA$^{tet}$ additions corresponded to some of the most perturbed peaks when 4.5S RNA was titrated previously [19]. The specific binding of PNA$^{tet}$ to the FtsY$_{NG}$ RNA-binding interface is further supported by partial assignments of backbone amide resonances of FtsY$_{NG}$ in the $^{15}$N-$^1$H-TROSY-HSQC spectrum. For example, S360, V403 and K405 on the RNA-binding site are some of the assigned residues that showed perturbations upon PNA$^{tet}$ and/or moRNA$^{tet}$ additions (**Fig 3F** and **3G**). Note that only ~70% of amide groups are visible on the $^{15}$N-$^1$H-TROSY-HSQC spectrum of FtsY$_{NG}$ presumably due to buried amide groups (produced as $^2$H$^{15}$N) have not been back-exchanged with $^1$H$^N$. Attempts to unfold and refold the protein in varying concentrations of urea were unsuccessful. Therefore, a more complete assignment of FtsY$_{NG}$ peaks could not be achieved and only unambiguous assignments are indicated in **S7A Fig in S1 File**. However, reassuringly, the assigned residues correspond mostly to surface residues, including some RNA-binding residues (**S7B Fig in S1**

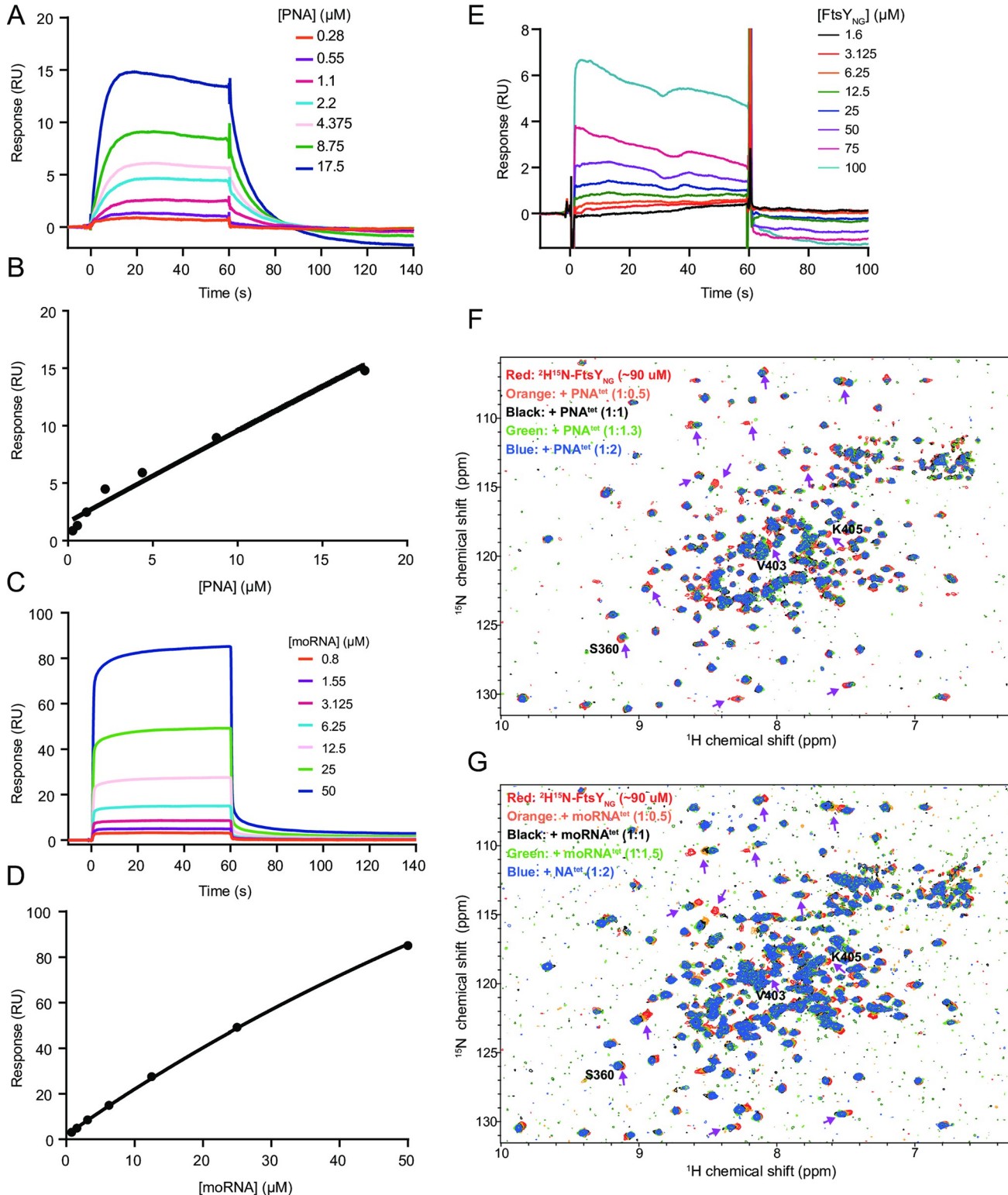

**Fig 3. Folded PNA^tet and moRNA^tet interact with FtsY_NG RNA-binding face in similar manners.** SPR sensorgram and fit to equilibrium response shown for PNA^tet (0.28–17.5 μM, A-B) or moRNA^tet (0.1–25 μM, C-D) binding to immobilised Avi-FtsY_NG. (E) SPR sensorgram of FtsY_NG (1.6–100 μM) binding to immobilised biotinylated PNA^tet. (F) $^{15}$N-$^{1}$H-TROSY-HSQC spectra of $^{2}$H$^{13}$C$^{15}$N-FtsY_NG alone (red) and following addition of 1:0.5, 1, 1.3 and 2 molar equivalence of PNA^tet (blue). (G) $^{15}$N-$^{1}$H-TROSY-HSQC spectra of $^{2}$H$^{13}$C$^{15}$N-FtsY_NG alone (red) and following addition of 1:0.5, 1, 1.5, and 2 molar equivalence of moRNA^tet (blue). Arrows indicate some of the perturbed residues.

File). $CA_n$ and $CA_{n-1}$ connectivities for the residues at the RNA-binding site in the HNCA are shown in **S8 Fig in S1 File**.

## PNA$^{tet}$ binds SARS-CoV-2 Nsp9

Having established that folded PNA$^{tet}$ can bind FtsY$_{NG}$ and disrupt the FtsY$_{NG}$:SRP interaction, we next wanted to know if PNAs can act as an RNA mimic for other RBPs. Nsp9 (non-structural protein 9) from SARS-CoV-2 was chosen as Nsp9 is a highly conserved and an essential component of the viral replication/transcription complex in coronaviruses [43, 44]. In addition, it is a dimeric protein that binds single-stranded RNA (ssRNA) non-specifically with weak affinity [23, 24]. However, recent results have revealed that it has a preference for binding hairpin structures [25] comparing to its weak affinity to ssRNA [23, 24, 44]. Therefore, we hypothesized that folded PNA$^{tet}$ might also bind Nsp9.

We have examined whether Nsp9 can bind to refolded PNA$^{tet}$ using SPR and NMR titration experiments. For SPR analysis, Nsp9 was immobilised, and increasing concentrations of PNA$^{tet}$ titrated. A binding response was observed (**Fig 4A**), however, a saturable binding curve was not observed over the assayable concentration range. Using IN3E3_LSA (an RNA hairpin designed to bind Nsp9 from [27], **S9 Fig in S1 File**) as a positive control, an estimate $K_D$ of $80 \pm 7$ μM was calculated for PNA$^{tet}$ (**Fig 4B**), which is ~5–10 fold higher than the $K_D$ for single-stranded RNA as previously published [23, 24] and from our own work [30].

To further delineate which Nsp9 residues are involved in binding PNA$^{tet}$, a $^{15}N$-$^{1}H$-TROSY-HSQC titration study was conducted using uniformly $^{15}N$-labelled Nsp9. Minor movement (as indicated in **Fig 4B**), and disappearance of several Nsp9 peaks were observed, confirming that PNA$^{tet}$ is indeed interacting with the protein. Comparison with the published data [30] on RNA binding of Nsp9 reveals that the binding interfaces of PNA and RNA overlap to some extent, suggesting that Nsp9 recognise PNA in a similar manner to ssRNA.

## Conclusion and discussion

PNA is a synthetic molecule that replaces ribose phosphate backbone of nucleic acid with polyamide, providing resistance to nucleases and protease attacks while retaining the hybridization property of base complementarity [45]. Compared with DNA and RNA, the Watson-Crick interactions in PNAs are not interfered by the electronegative repulsion from the backbone, and this property has been exploited to interfere with native base pairing. In addition, the biostability of PNA makes it a promising therapeutic agent and it has been used as an antisense inhibitor to control gene/protein expression *in vivo* [46]. However, previous studies have only explored the base complementarity approach to inhibit transcription or block translation [47–49]. In this study, we have shown engineered PNAs that mimic the 4.5S tetraloop and can fold into a hairpin structure despite the differences in the backbone between RNA and PNA. In addition, the folded PNA$^{tet}$ has been shown to bind two RBPs. Encouragingly, despite a 3-fold reduction in binding affinity, PNA$^{tet}$ can readily compete with the native 4.5S RNA and prevent SRP:FtsY$_{NG}$ complex formation benchmarking our new concept that PNAs can be used to abrogate RBP:RNA interactions in assays, especially those that would be difficult to carry out under RNase-free conditions. The ability for PNA$^{tet}$ to compete for 4.5S RNA despite its lower binding affinity possibly result from its more robust hairpin structure. However, for PNAs to become useful tool compounds and drugs, further developments are needed to improve the solubility, affinity, specificity and cell permeability.

The MST results indicated that PNA with longer stem, such as PNA$^{tetL}$, is more effective in terms of interfere FtsY-RNA interaction. However, PNAs intrinsically have low aqueous solubility due to the charge-neutral backbone and the solubility of PNA tends to decrease as the

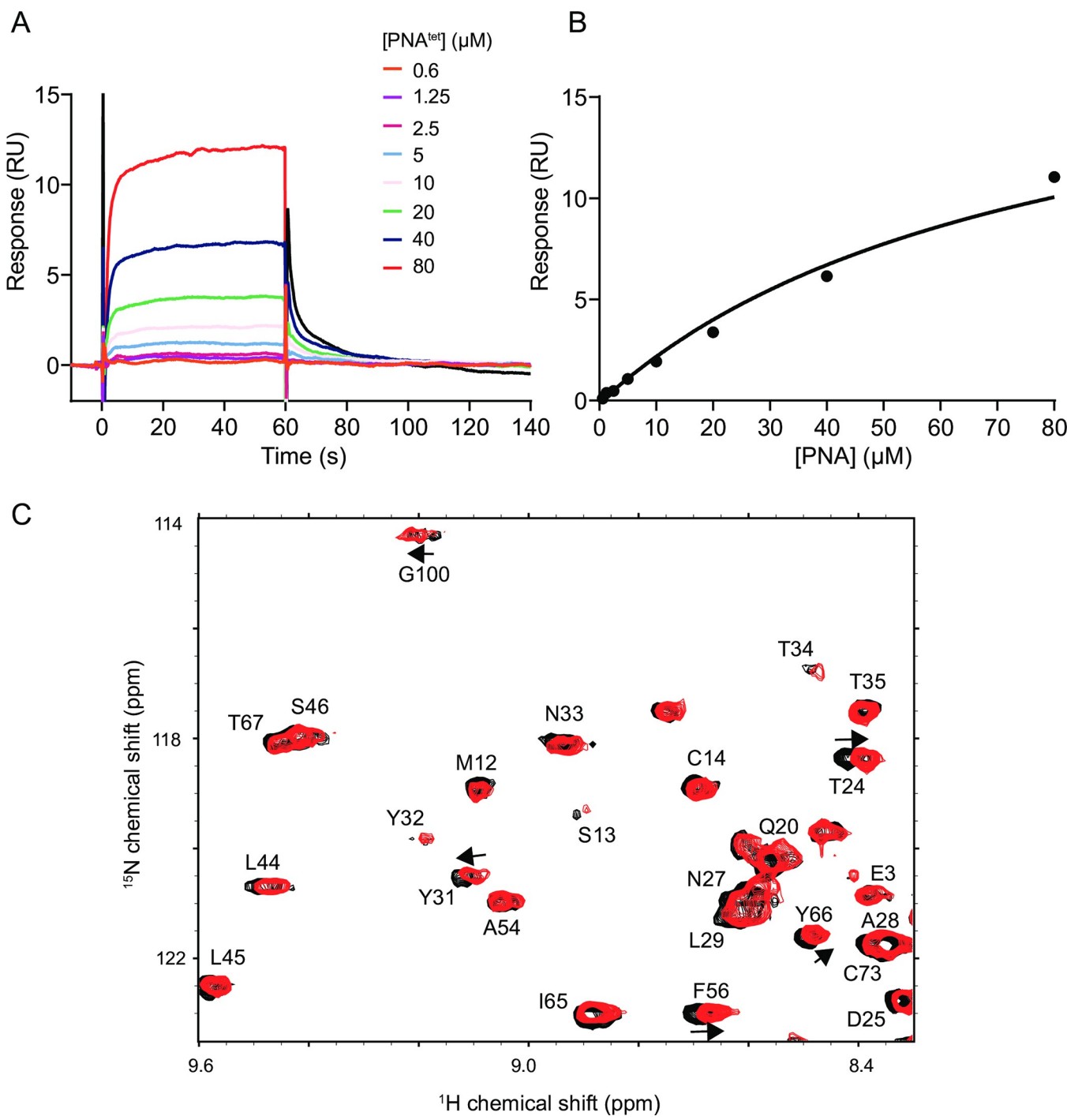

**Fig 4. PNAtet showing binding to Nsp9.** (A-B) Sensorgram and fit to equilibrium response shown for PNAtet (0.6–80 μM) binding to immobilised Avi-Nsp9. (C) 15N-1H-TROSY-HSQC spectra of 15N-Nsp9 alone (blue) and following addition of 1:1 molar equivalence of PNAtet (red). Arrows indicate some of the perturbed residues.

oligomer length (e.g., a longer stem) and purine:pyrimidine ratio increase [50] and this was also observed with the tested PNAs. Thus, the effectiveness of PNA inhibition can likely benefit from improving the solubility of longer PNA molecules. Indeed, backbone modifications as well as internal or terminal linkers have been successfully used to improve PNA aqueous

solubility. Unmodified PNA molecules have an achiral polyamide backbone consisting of *N*-(2 aminoethyl) glycine (AEG). The glycyl α carbon can be linked to the acetamido β carbon on the same PNA unit via a methylene bridge, creating a *N*-(2 aminoethyl) prolyl (AEP) backbone, thus increasing water solubility by introducing positive charges [51]. Similarly, adding a hydrophilic (R)-diethylene glycol [52] or lysine side chain [53] to the γ carbon of the AEG backbone can also enhance solubility and reduce PNA self-aggregation. However, backbone modifications have the drawback of increasing conformational rigidity and/or chirality to the molecule, which may disrupt PNA folding. Therefore, a more common approach is to include terminal tags or internal linkers, which improve aqueous solubility by adding hydrophilic moieties. Such as the AEEA-linker used in this study, as well as X or E linkers, which are PNA units with nucleotide bases replaced by branched carboxylic acid derivatives. Since most cellular experiments only require PNA at micromolar to nanomolar concentrations, these modifications offer sufficient solubility enhancement.

Due to the chemical properties of PNAs, not all RBPs are suitable targets. While the addition of glutamic acid side chain on the gamma carbon of Gs and Ts in PNAs can compensate for the negative charge in the ribose-phosphate backbone, RBPs that bind RNAs with a very high affinity and mainly through interactions with the backbone are unsuitable for PNA targeting. As shown in our study, RBPs that bind to nucleobases with structure and sequence specificity at low to moderate affinities (in the μM range) can likely be targeted by PNAs. In this study, we based our PNA design on the native 4.5S RNA tetraloop to target the FtsY. While PNA$^{tet}$ and variants bind FtsY$_{NG}$ and displays inhibitory activity in competition assays *in vitro*, it is possible that the native GNRA tetraloop sequence might not be the best sequence for the PNA design. Systematic evolution of ligands by exponential enrichment (SELEX) experiments have demonstrated for many RBPs that evolved sequences often surpass the native binding sequence in binding affinity and selectivity [54, 55]. The RNA-binding UP1 domain of heterogeneous nuclear ribonucleoproteins, for example, specifically recognises a native 5′- UUAGGG -3′ sequence on pre-mRNA for 3′ splicing [56, 57]. However, high-throughput affinity analysis studies showed a strong affinity to 5′- GUAGGAG -3′ sequence [58]. Affinity distribution analyses, which analyse the RNA association kinetics and affinity for all possible sequence combinations of a given length have been successfully used to identify "hidden" binding sequences. For instance, the C5 subunit of *Escherichia coli* RNase P does not have a known consensus sequence but its affinity distribution is similar to those of specific RBPs, indicating that C5 has inherent specificity [59]. Therefore, if higher affinity and specificity is required for a particular application, PNA design can include an optimised RNA sequence for the nucleobases while maintaining the advantages of more stable base pairing, due to the lack of backbone repulsion [60]. Furthermore, our results demonstrated that the refolded PNAs were highly stable structurally and could maintain hairpin structures after freeze-thawing as well as freeze-drying as well as reconstitution and long-term storage in aqueous solutions.

In conclusion, our results benchmark the potential of PNAs as a new class of compounds that can be used to specifically target the RNA-binding sites of RBPs in biochemical and biophysical assays, especially those impractical to perform under RNase-free conditions. Together with improvements in PNA solubility, cell permeability and structure-activity relationship that are under development, modified PNA molecules may also have potential to be developed into new drugs from anti-microbials to anti-cancer agents given the importance of RBPs:nucleobase interactions in biology. In addition, folded PNA may offer additional advantages as it can prevent off-target effects that arise from non-intentional base complementarity with endogenous RNA and DNA sequences *in vivo*.

## Supporting information

**S1 Raw images. This file includes all raw gel images.**
(PDF)

**S1 File. This file includes S1 to S9 Figs with legends.**
(PDF)

**S2 File. This folder includes all raw files in folders grouped by methods.**
(ZIP)

**S3 File. This file includes a description of the supplementary data/files for figures.**
(DOCX)

## Acknowledgments

The authors acknowledge the Sydney Analytical Core Research Facility for access to SPR and NMR infrastructure and Rezwan Siddiquee for assisting with FtsY$_{NG}$ purification.

## Author Contributions

**Conceptualization:** Yichen Zhong, Roland Gamsjaeger, Sandro F. Ataide, Ann H. Kwan.

**Formal analysis:** Yichen Zhong, Esther Zhang, Biswaranjan Mohanty, Belinda B. Zhang, Madeline S. McRae, Roland Gamsjaeger, Sandro F. Ataide, Ann H. Kwan.

**Funding acquisition:** Sandro F. Ataide, Ann H. Kwan.

**Investigation:** Yichen Zhong, Lorna Wilkinson-White, Esther Zhang, Biswaranjan Mohanty, Belinda B. Zhang, Madeline S. McRae, Rachel Luo, Thomas A. Allport, Anthony P. Duff, Jennifer Zhao, Liza Cubeddu, Roland Gamsjaeger, Sandro F. Ataide, Ann H. Kwan.

**Methodology:** Yichen Zhong, Lorna Wilkinson-White, Esther Zhang, Biswaranjan Mohanty, Belinda B. Zhang, Madeline S. McRae, Rachel Luo, Thomas A. Allport, Anthony P. Duff, Jennifer Zhao, Serene El-Kamand, Mar-Dean Du Plessis, Roland Gamsjaeger, Sandro F. Ataide, Ann H. Kwan.

**Project administration:** Ann H. Kwan.

**Supervision:** Sandro F. Ataide, Ann H. Kwan.

**Writing – original draft:** Yichen Zhong, Lorna Wilkinson-White, Esther Zhang, Biswaranjan Mohanty, Belinda B. Zhang, Roland Gamsjaeger, Sandro F. Ataide, Ann H. Kwan.

**Writing – review & editing:** Yichen Zhong, Lorna Wilkinson-White, Biswaranjan Mohanty, Anthony P. Duff, Roland Gamsjaeger, Sandro F. Ataide, Ann H. Kwan.

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
