## [Decision Letter · Decision Letter 0]

6 Aug 2024

PONE-D-24-28006Peptide Nucleic Acids can form hairpins and bind RNA-binding proteinsPLOS ONE

Dear Dr. Kwan,

Thank you for submitting your manuscript to PLOS ONE. After careful consideration, we feel that it has merit but does not fully meet PLOS ONE’s publication criteria as it currently stands. Therefore, we invite you to submit a revised version of the manuscript that addresses the points raised during the review process.

As you will see one Reviewer suggested minor revisions, while the other one suggested a major revision. However, based on the comments from both Reviewers I think that a minor revision is the most appropriate choice. Reviewer 1, who is an expert in RNA-ligand and protein-ligand interaction, as well as in structural biology, is extremely satisfied with the set of experiments used to prove most of your claims. Nonetheless, I will ask you to correct the manuscript by addressing all of their concerns:

1. Add a discussion of the future directions of how to improve the PNA/protein binding strength and specificity. In particular, what kind of PNA SL modifications: structural, chemical, sequence (?) could make it a better competitor for the cognate RNA SL?

2. It is not clear what role glutamate has in binding, if any at all. Please explain it.

3. What is the OO linker?

4. Fig 2A: why are the three black lines different in the three panels?

5. Line 472: It is not clear what the authors mean by “using GDP binding as a positive control”. Do they mean competitive inhibitor?

6. The RNA-FtsY interaction is well-known to be a transient interaction. Does the PNA have any meaningful inhibitory role in a functional SRP assay (GTP hydrolysis, protein targeting, secretion etc?)

Reviewer 2 also suggested to compare one PNA without glutamate to assess this. I would strongly encourage you to perform this experiment; however, if not possible please justify it. Also, I would ask you to address to the best of your abilities if Ffh-RNA binding can be similarly inhibited by the synthesized PNA. It would be ideal to have experimental data to support this answer.

Al the best,

We look forward to receiving your revised manuscript.

Kind regards,

Mauricio Comas-Garcia

Academic Editor

PLOS ONE

Journal Requirements:

2. We note that this submission includes NMR spectroscopy data. We would recommend that you include the following information in your methods section or as Supporting Information files:

a) The make/source of the NMR instrument used in your study, as well as the magnetic field strength. For each individual experiment, please also list: the nucleus being measured; the sample concentration; the solvent in which the sample is dissolved and if solvent signal suppression was used; the reference standard and the temperature.

b) A list of the chemical shifts for all compounds characterised by NMR spectroscopy, specifying, where relevant: the chemical shift (δ), the multiplicity and the coupling constants (in Hz), for the appropriate nuclei used for assignment.

c)The full integrated NMR spectrum, clearly labelled with the compound name and chemical structure.

We also strongly encourage authors to provide primary NMR data files, in particular for new compounds which have not been characterised in the existing literature. Authors should provide the acquisition data, FID files and processing parameters for each experiment, clearly labelled with the compound name and identifier, as well as a structure file for each provided dataset. See our list of recommended repositories here: https://journals.plos.org/plosone/s/recommended-repositories

   "The University of Sydney Drug Discovery Initiative (DDI) seed funding 

The production of 2H13C15N FtsYNG was supported by grant NDF9615 from the National Deuteration Facility, which is partly supported by the National Collaborative Research Infrastructure Strategy – an initiative of the Australian Government."

5. In this instance it seems there may be acceptable restrictions in place that prevent the public sharing of your minimal data. However, in line with our goal of ensuring long-term data availability to all interested researchers, PLOS’ Data Policy states that authors cannot be the sole named individuals responsible for ensuring data access (http://journals.plos.org/plosone/s/data-availability#loc-acceptable-data-sharing-methods).

7. PLOS ONE now requires that authors provide the original uncropped and unadjusted images underlying all blot or gel results reported in a submission’s figures or Supporting Information files. This policy and the journal’s other requirements for blot/gel reporting and figure preparation are described in detail at https://journals.plos.org/plosone/s/figures#loc-blot-and-gel-reporting-requirements and https://journals.plos.org/plosone/s/figures#loc-preparing-figures-from-image-files. When you submit your revised manuscript, please ensure that your figures adhere fully to these guidelines and provide the original underlying images for all blot or gel data reported in your submission. See the following link for instructions on providing the original image data: https://journals.plos.org/plosone/s/figures#loc-original-images-for-blots-and-gels.   

Additional Editor Comments:

Dear Prof. Kwan,

I want to thank your for your submission. As you will see one Reviewer suggested minor revisions, while the other one suggested a major revision. However, based on the comments from both Reviewers I think that a minor revision is the most appropriate choice. Reviewer 1, who is an expert in RNA-ligand and protein-ligand interaction, as well as in structural biology, is extremely satisfied with the set of experiments used to prove most of your claims. Nonetheless, I will ask you to correct the manuscript by addressing all of their concerns:

1. Add a discussion of the future directions of how to improve the PNA/protein binding strength and specificity. In particular, what kind of PNA SL modifications: structural, chemical, sequence (?) could make it a better competitor for the cognate RNA SL?

2. It is not clear what role glutamate has in binding, if any at all. Please explain it.

3. What is the OO linker?

4. Fig 2A: why are the three black lines different in the three panels?

5. Line 472: It is not clear what the authors mean by “using GDP binding as a positive control”. Do they mean competitive inhibitor?

6. The RNA-FtsY interaction is well-known to be a transient interaction. Does the PNA have any meaningful inhibitory role in a functional SRP assay (GTP hydrolysis, protein targeting, secretion etc?)

Reviewer 2 also suggested to compare one PNA without glutamate to assess this. I would strongly encourage you to perform this experiment; however, if not possible please justify it. Also, I would ask you to address to the best of your abilities if Ffh-RNA binding can be similarly inhibited by the synthesized PNA. It would be ideal to have experimental data to support this answer.

Al the best,

Prof. Mauricio Comas-Garcia

Handling Editor

Reviewers' comments:

Reviewer's Responses to Questions

**Comments to the Author**

1. Is the manuscript technically sound, and do the data support the conclusions?

Reviewer #1: Yes

Reviewer #2: Partly

2. Has the statistical analysis been performed appropriately and rigorously? 

Reviewer #1: Yes

Reviewer #2: I Don't Know

3. Have the authors made all data underlying the findings in their manuscript fully available?

Reviewer #1: Yes

Reviewer #2: Yes

4. Is the manuscript presented in an intelligible fashion and written in standard English?

Reviewer #1: Yes

Reviewer #2: Yes

5. Review Comments to the Author

Reviewer #1: Review in the manuscript “Peptide Nucleic Acids can form hairpins and bind RNA-binding proteins” by Yichen Zhong et.al.

In this manuscript the authors explore an innovative idea that the peptide nucleic acids (PNAs) can be used as a drug that will compete with the RNA for its functional partner protein binding. They hypothesize that the short PNA molecule of the same sequence as prototype RNA can fold into the same secondary structure and bind the same protein partner via the same interface. This is far from obvious, and the authors managed to prove this hypothesis for the short stem loop (SL) structure by using the impressive array of experimental approaches, including NMR, SPR, CD, EMSA, and MST. This work is convincing as a proof of principle. The authors have shown quite convincingly by a combination of approaches that their PNA SL folds in the same hairpin structure as analogous RNA and binds protein with the same surface. The big advantage of the PNA vs RNA drug is that PNA is not degradable by the cellular nucleases.

However, the efficiency of such PNA mimic of RNA SL in competing for its partner protein at least in the case studied by authors is relatively low. Thus, the competition binding experiments presented in Fig. 2 suggest that even the best competitor PNAtelL of the RNA SL for binding the partner protein I only weakly effective. Indeed, addition of 5 �M of this competitor PNA SL to the protein makes the cognate RNA SL bind to it only ~2.5-fold weaker, with Kd changing from ~100nM to ~300nM. Based on that result we can roughly estimate that the PNAtelL Kd for that protein is ~50-fold weaker that the RNA Kd for the same protein. However, maybe the PNA can be further modified to improve its cognate protein binding. This seems possible, as according to the same competition studies, the strength of PNA/protein interaction is strongly affected by the length of the PNA stem, the sequence of its loop, type of the PNA or modified RNA backbone used. I believe this paper would benefit from discussion of the future directions of how to improve the PNA/protein binding strength and specificity. What kind of PNA SL modifications: structural, chemical, sequence (?) could make it a better competitor for the cognate RNA SL?

Reviewer #2: This communication by Kwan and coworkers reports the use of peptide nucleic acids (PNAs) as RNA mimics for competitive inhibition of RNA-protein interactions. The authors have demonstrated this with the help of two examples, including that of SRP inhibition.

Overall, the text is well written and is worthy of publication. However, there are a few queries that need to be carefully addressed before the paper can be considered suitable for publication.

It is not clear what role glutamate has in binding, if any at all. The authors should compare one PNA without glutamate to assess this.

What is OO linker?

Fig 2A: why are the three black lines different in the three panels? I understand that they are all measuring RNA-FtsY binding in the absence of PNA. As Ffh is also well-known to bind to RNA, the authors must check whether Ffh-RNA binding can be similarly inhibited by the synthesized PNA.

Lin2 472: It is not clear what the authors mean by “using GDP binding as a positive control”. Do they mean competitive inhibitor?

The RNA-FtsY interaction is well-known to be a transient interaction. Does the PNA have any meaningful inhibitory role in a functional SRP assay (GTP hydrolysis, protein targeting, secretion etc?)

6. PLOS authors have the option to publish the peer review history of their article (what does this mean?). If published, this will include your full peer review and any attached files.

Reviewer #1: **Yes: **Ioulia Rouzina

Reviewer #2: No

---

## [Author Response · Author response to Decision Letter 0]

25 Aug 2024

Editorial comments summarising comments from Reviewer 1 and 2:

Reviewer 1’s comments:

1. Add a discussion of the future directions of how to improve the PNA/protein binding strength and specificity. In particular, what kind of PNA SL modifications: structural, chemical, sequence (?) could make it a better competitor for the cognate RNA SL?

We agree with the reviewer that the strength of PNA/protein interaction is strongly affected by the length of the PNA stem, the sequence of its loop, type of the PNA or modifications used. The longer PNA performed better in competition assays, but at the cost of solubility. We have expanded on the discussion about general strategies to improve PNA solubility in the manuscript text. Please refer to the section from Lines 547 to 565.

The other strategy to improve binding affinity is by optimising the binding sequence. In Lines 573 to 586, we describe how SELEX and affinity distribution analysis may be used to find better binders. Another way to improve the PNA-Protein binding affinity is via rational design strategies using high resolution structures of the PNA-protein complex. We have attempted to crystalise the complex but have not had success so far and we suspect this is at least partially due to the limited PNA solubility and the requirement for a high concentration of PNA-protein to form crystals. It is very possible that further chemical modifications to the PNA backbone to allow H-bonding with the protein and expanding those interactions with additional residues can help. Unfortunately for this study, we are limited to a small number of commercially available modifications even though PNA modifications to enhance solubility and functions are a hot area of research with several recent publications. We are hopeful that our findings would expand the interest and demand for PNAs and this would further drive research in PNA modification and synthesis. We have recently set up a collaboration with Dr Emma Watson, University of Adelaide, who has embarked on improving PNA capabilities in her new laboratory following on from her postdoctoral work:

https://onlinelibrary.wiley.com/doi/full/10.1002/hlca.202300110

Reviewer 2’s comments:

2. It is not clear what role glutamate has in binding, if any at all. Please explain it.

The glutamate was added to the PNA at the tetraloop to mimic the phosphate from the 4.5SRNA backbone that interacts with a positive patch of amino acids from the FtsY. To test the effect of this glutamate, we have used a PNAtet variant without glutamate in MST and SPR assays. However, this variant aggregated on the SPR surface and produced noisy and uninterpretable MST results. We also tested a glutamate-free variant of PNAtetN and observed similar results. It is likely this glutamate also serves as a solubility and/or stability modification to the PNA. Note that PNAtetN, which retains the glutamate but with a different tetraloop sequence did not inhibit the FtsY and RNA interaction, indicating that the loop sequence is the main contributor.

3. What is the OO linker?

The OO linker is a 2-aminoethoxy-2-ethoxy acetic acid (AEEA) linker. OO linker is the name used by the manufacturer, and we have changed it to AEEA linker in the revised manuscript. It was added to increase the solubility of PNA.

4. Fig 2A: why are the three black lines different in the three panels?

The variations in KD observed are likely due to different batches of purified protein and/or RNA. In particular, the fluorescently labelled RNA degrades easily, and this is a major motivation for using PNAs as RNA mimics. The KD between FtsY and RNA is typically ~100−200 �M , but occasionally it has been observed to increase to ~300 �M like in panel 3. The experiments reported in panel 3 were performed a few months after those shown in the first two panels as the different PNAs were ordered and available at different times. Therefore, we have chosen to compare relative rather than absolute binding affinities in the MST assays. In this study, the affinities measured in the presence and absence of PNAs were performed on the same day using the same aliquot of protein and RNA and the ratio of affinity was presented.

Additional comment from Review 2 following on the above question

As Ffh is also well-known to bind to RNA, the authors must check whether Ffh-RNA binding can be similarly inhibited by the synthesized PNA.

PNAtet and its variants would not be able to inhibit the Ffh-4.5S RNA interaction because that interaction is much stronger and is mediated via the M domain and not the S-domain tetraloop. The M domain that binds Ffh is an asymmetrical loop located in another part of the stem and is surrounded by a different structure and sequence.

5. Line 472: It is not clear what the authors mean by “using GDP binding as a positive control”. Do they mean competitive inhibitor?

Due to the weak nature of the FtsYNG:PNA interactions and limited solubility of the PNA, the binding responses in SPR do not go to saturation even at the highest PNA concentrations used. Therefore, the Rmax parameter for the PNA binding was fixed during the curve fitting process and as estimated using the fitted maximum response for the positive control (GDP). We have modified the manuscript text to reflect this and added a reference in Line 477 to our previous paper [1] and ref 19 in manuscript main text.

6. The RNA-FtsY interaction is well-known to be a transient interaction. Does the PNA have any meaningful inhibitory role in a functional SRP assay (GTP hydrolysis, protein targeting, secretion etc?)

Given the designed PNA can inhibit the initial step of complex formation according to EMSA, we would expect inhibition of all subsequent functional steps but this remains to be tested. We are planning to carry out more assays to investigate the functions of the PNA on SRP assembly and its biological effects including in vivo activity. However, these are beyond the scope of the current study which provides proof-of-concept that PNAs can be used to specifically target the RNA-binding sites of RBPs in biochemical and biophysical assays.

1. Faoro C, Wilkinson-White L, Kwan AH, Ataide SF. Discovery of fragments that target key interactions in the signal recognition particle (SRP) as potential leads for a new class of antibiotics. PLoS One. 2018;13(7):e0200387.

---

## [Editor Report · Decision Letter 1]

30 Aug 2024

Peptide Nucleic Acids can form hairpins and bind RNA-binding proteins

PONE-D-24-28006R1

Dear Dr. Kwan,

We’re pleased to inform you that your manuscript has been judged scientifically suitable for publication and will be formally accepted for publication once it meets all outstanding technical requirements.

Kind regards,

Mauricio Comas-Garcia

Academic Editor

PLOS ONE

---

## [Editor Report · Acceptance letter]

5 Sep 2024

PONE-D-24-28006R1 

PLOS ONE

Dear Dr. Kwan, 

I'm pleased to inform you that your manuscript has been deemed suitable for publication in PLOS ONE. Congratulations! Your manuscript is now being handed over to our production team.

Kind regards, 

on behalf of

Dr. Mauricio Comas-Garcia 

Academic Editor

PLOS ONE